# The Acrylamide Degradation by Probiotic Strain *Lactobacillus acidophilus* LA-5

**DOI:** 10.3390/foods11030365

**Published:** 2022-01-27

**Authors:** Katarzyna Petka, Paweł Sroka, Tomasz Tarko, Aleksandra Duda-Chodak

**Affiliations:** 1Department of Plant Products Technology and Nutrition Hygiene, Faculty of Food Technology, University of Agriculture in Krakow, 30-149 Kraków, Poland; katarzyna.petka@urk.edu.pl; 2Department of Fermentation Technology and Microbiology, Faculty of Food Technology, University of Agriculture in Krakow, 30-149 Kraków, Poland; pawel.sroka@urk.edu.pl (P.S.); tomasz.tarko@urk.edu.pl (T.T.)

**Keywords:** acrylamide, lactic acid fermentation, probiotics

## Abstract

Acrylamide is a harmful substance produced in thermal processed food; however, it can also be found in food with various additives. The aim of the study was to check whether the probiotic bacteria strain, *Lactobacillus acidophilus* LA-5 (LA5), can degrade acrylamide and hence reduce its concentration in foodstuff. Our results revealed that LA5 can degrade acrylamide and cause a decrease in its concentration, but only when other available carbon and nitrogen sources are lacking. In the presence of casein, lactose, milk fat or in whole cow’s milk, this ability disappeared. Acrylamide present in milk, however, modulated the bacteria metabolism by significantly enhancing lactic acid production by LA5 in milk (at conc. 100 µg/mL), while the production of acetic acid was rather reduced.

## 1. Introduction

Acrylamide (AA) is a harmful substance, found not only in the environment, but also in food products. Moreover, in 1994, acrylamide was classified as “probably cancerogenic” [1]. AA can be formed in food during thermal treatment at temperatures above 120 °C, mainly as a result of the Maillard reaction between asparagine and reducing sugars, by the so-called acrolein pathway or during the thermal degradation of gluten. It may also be of endogenous origin, as, under the conditions of physiological temperature, pH, and ionic strength, AA can be formed from asparagine under the influence of reactive oxygen species, which are generated under various pathological conditions with oxidative stress [2]. However, AA can also be present in non-heat-treated food, as it is added to fermented milk or yoghurts with products such as muesli, roasted nuts, high-temperature dried fruit, seeds, coffee, etc., usually after the fermentation process.

Due to its negative impact on human health, studies focused on the reduction in AA concentration in food have been conducted for many years. Among the most common mitigation strategies are [3]: choosing varieties of food raw material with reduced asparagine and/or glucose content, removing the AA precursors from the food before its thermal treatment, the enzymatic conversion of asparagine and glucose to aspartic acid (e.g., by using L-asparaginase), the selection of appropriate processing conditions and techniques (time, temperature, vacuum, blanching, fermentation etc.), and the introduction of appropriate food additives (e.g., antioxidants). All of them cause reduced AA formation during food processing, but they do not show how to reduce the concentration of AA that was already produced.

Some studies showed the ability of soil and water bacteria to degrade AA by producing enzymes belonging to amidases [4]. It should be noted that some bacteria living in our digestive tract can also produce amidases [5]. The research of Serrano-Niño et al. [6] reported that some species of lactic acid bacteria (LAB), especially those from *Lactobacillus* and *Bifidobacterium* genera, can remove various toxins and carcinogens (including AA) in vitro, as well as ex vivo and in vivo [6,7].

Probiotics, according to FAO and WHO definition, are live microorganisms that, when administered in adequate amounts, confer a health benefit to the host” [8]. *Lactobacillus acidophilus* LA-5 (LA5) is one of the probiotic strains that is most often added to fermented milk drinks. *L. acidophilus* was also identified as part of the human microbiota in oral, digestive, and vaginal tracts [9,10]. This is important, because LAB, which are present in fermented milk drinks or in the gut, are still alive, so they need nutrients, and it is possible that AA can be used by them while still present in food. Recently, *L. acidophilus,* as a probiotic strain, is increasingly used in the production of bread with pro-health properties. Cizeikiene et al. [11] proved that *L. acidophilus* can be used to obtain fermented quinoa, which can then be used as a non-traditional sourdough for the production of quinoa–wheat composite bread, characterised by an increased biological value as well as improved acceptability and sensory properties. Moreover, there are some studies proving that probiotic *L. acidophilus* can survive in bread during baking and storage [12].

In previous studies [5,13], it was proved that the growth in *L. acidophilus* LA-5 in model medium (a medium with limited accessibility to carbon and nitrogen sources) was stimulated by acrylamide at a concentration of 7.5 µg/mL, and this was probably due to the this bacteria strain’s ability to create the decomposition of AA by amidase. The ammonia released by bacteria amidase was then used by bacteria as a source of nitrogen, while the rest of the molecule (probably acrylic acid or its metabolites) was used as a carbon source.

Based on the current state of knowledge and the results of previous research [5,13], a hypothesis that “the probiotic bacteria strain, *Lactobacillus acidophilus* LA-5, can degrade acrylamide and reduce its concentration in the environment” was formulated. To verify this hypothesis, a series of subsequent experiments was conducted, aiming to check whether *L. acidophilus* LA-5 can decrease the acrylamide concentration in various environments (with limited carbon and nitrogen sources or in real “food” conditions). Moreover, we also checked whether AA can influence the bacteria metabolism when it is present in milk.

## 2. Materials and Methods

### 2.1. Chemicals

Acrylamide (purum, ≥98%, GC), methacrylamide (purum, >98%, GC), acetonitrile (for HPLC), as well as standards for HPLC (glucose, lactose, lactic acid and acetic acid) were provided by Sigma-Aldrich (Sp. z o.o, Poznan, Poland). Activated charcoal (for analysis) and silica gel 60 (0.040–0.063 mm, for column chromatography) were obtained from Merck Sp. z o.o. (Warsaw, Poland), while methanol (HPLC-suitable), sodium chloride, ethyl acetate, sulphuric acid, and sodium sulphate anhydrous (all pure p.a.) from Avantor Performance Materials Poland S.A. Lactose, as well as Carrez’s Reagent I (1 mol/L of zinc acetate in 0.5 mol/L acetic acid solution) and Carrez’s Reagent II (aqueous 0.25 mol/L solution of potassium hexacyanoferrate(II)), used for precipitation and clarification, were provided by Chempur (Piekary Śląskie, Poland). Casein was obtained from Pol-Aura, Chemicals (Olsztyn, Poland).

### 2.2. Bacteria Strain and Microbial Media

Probiotic strain *Lactobacillus acidophilus* LA-5 was provided by Chr. Hanses (Hørsholm, Denmark).

In the study, an MRS (de Man, Rogosa and Sharpe Broth, BioMaxima, Lublin, Poland) medium was used for bacteria propagation, while a Maximum Recovery Diluent (MRD) was used as a model medium with limited accessibility to carbon and nitrogen compounds. MRD was composed of 0.45% NaCl (*w*/*v*) and 0.45% (*w*/*v*) of bacteriological peptone (BioMaxima), which was the only source of carbon and nitrogen in this medium. The mean pH of sterile MRD was 6.6.

If solid medium was required, a bacteriological agar was added at a final concentration of 2% (BioMaxima, Lublin, Poland). All media were sterilised using a Microjet Microwave Autoclave (Enbio Technology Sp. z o.o., Gdynia, Poland) and the parameters of the sterilisation process were 80 s, 135 °C, and 3.6 bar.

### 2.3. Bacteria Preparation

The bacteria were delivered as a freeze-dried culture, and were rehydrated according to the supplier’s protocol. The bacteria were incubated at an optimal temperature, i.e., at 37 °C.

Fresh culture was prepared before each experiment, i.e., bacteria were cultured for 24 h in MRS broth at 37 °C. Just before the experiments, the bacteria culture was centrifuged (194× *g* for 15 min, MPW-35JR centrifuge, MPW MED Instruments, Warsaw, Poland). The bacteria were rinsed twice with sterile distilled water to remove the growth medium residues, and the pellet was resuspended in MRD to obtain bacterial suspensions. The bacteria concentration was adjusted to 10^7^ cfu/mL using the optical density measurement for this purpose. The bacteria concentration was calculated from the relationship established during calibration, as described below.

### 2.4. Measurement of Optical Density of Bacterial Suspension: Calibration

A volume of 0.1 mL of 24-h liquid *Lactobacillus acidophilus* LA-5 culture was added to a tube containing 5 mL of sterile MRS medium, the content was mixed, and the tube was incubated for 24 h at 37 °C. After incubation, the culture was centrifuged (the parameters used as in the chapter above) and cells were rinsed, as described in the previous chapter. The resulting pellet was resuspended in sterile water to obtain the optical density of bacterial suspension equal to the McFarland standard 1.0 (using a Den-1B densitometer, Biosan, Latvia). Then, serial 10-fold dilutions were made in MRD, and 1 mL of subsequent dilution was inoculated in the MRS medium using the pour plate method (each dilution was inoculated in triplicate). After 72 h of incubation at 37 °C in aerobic conditions, bacterial colonies were counted, and the mean bacterial cell density (expressed in cfu/mL) from the replicates was calculated. The relationship between the optical density in McFarland units and bacterial cell density was determined. For *L. acidophilus* LA-5, the optical density value in McFarland units that corresponds to the cell concentration of 10^7^ cfu/mL was 0.22.

### 2.5. Preparation of Acrylamide “Stock” Solution

A concentrated (1 g/L) aqueous solution of acrylamide was sterilised by membrane filtration (pore φ = 0.22 µm; PES Millex-GP, Bionovo, Poland) and diluted (if needed) with sterile distilled water to obtain “Stock” solutions of acrylamide (concentrations: 75.0, 150.0, 300.0, and 1000 mg/L). Solutions were kept in refrigerated (4 °C) and dark conditions before experiments.

### 2.6. The Evaluation of Acrylamide Degradation by Lactobacillus acidophilus LA-5

#### 2.6.1. The Degradation of Acrylamide in a Model Medium

At the beginning, we checked whether *Lactobacillus acidophilus* LA-5 was able to degrade acrylamide in limited carbon- and nitrogen- conditions and the concentration of acrylamide actually decreased in the medium. For that purpose, 1 mL of adequate acrylamide “Stock” solution was added to 8 mL of MRD (giving the final concentrations of 7.5, 15, 30, or 100 µg/mL) and samples were inoculated with 1 mL of *Lactobacillus acidophilus* LA-5 suspension (prepared as described in Section 2.3). At the beginning (time 0 h), and after 24 h and 48 h of incubation at 37 °C, samples were centrifuged at 3000× *g* for 3 min and the supernatant was collected for the assessment of acrylamide concentration by gas chromatography. Separate sets of experimental variants were prepared for each time interval (0 h, 24 h and 48 h). The positive controls contained only MRD medium and adequate acrylamide “Stock” solution (concentration from 7.5 to 100 µg/mL), without the addition of bacteria. The negative controls were tubes containing MRD medium inoculated with bacteria suspension (as above), but without acrylamide addition (1 mL of sterile water instead). The concentration of acrylamide in test samples was compared to the acrylamide concentration in controls (samples without bacteria) to prove that the acrylamide degradation was dependent on microorganisms. The experiment was performed in six replicates.

Adequate cell-free controls and blanks were also included in the experiments to prove that acrylamide detected in the samples was not a residue from any laboratory ware, nor it was formed as an artefact in any analytical procedure. Controls confirmed that the samples did not contain any measurable amounts of acrylamide, except those samples where AA was added by authors. To prove that the decreased concentration of acrylamide was not a result of its binding to bacteria cells, trials with 10-fold higher amounts of bacteria were performed.

#### 2.6.2. The Impact of Basic Milk Components on the Degradation of Acrylamide

In the next step, we checked whether the ability of bacteria to degrade acrylamide was maintained in the presence of food matrix components that include easily available nutrients. As *Lactobacillus acidophils* LA-5 is usually used in yoghurts, we checked how the addition of basic milk components to the MRD medium influenced the ability of the tested bacteria to acrylamide degradation. For that purpose, the following components were added to the MRD medium: (1) the main milk protein, casein (at concentrations 2.5%, 2.6%, 2.7%, and 2.8%), (2) the main milk sugar, lactose (at concentrations 4.6%, 4.7%, 4.8%, 4.9%, and 5.0%), or (3) a milk fat, butter (at concentrations 3.6%, 4.0%, 4.3%, and 4.6%). The concentrations were selected based on the mean values reported for cow milk [14].

The experiment was carried out similarly to the above description, with the difference being that, instead of a pure MRD medium, an MRD with the addition of adequate concentrations of casein, lactose or milk fat was used. The supernatants were collected for the assessment of acrylamide concentration using gas chromatography. For each timepoint, separate sets of experimental variants were prepared. Each experimental sample or control variant (with/without milk component, with/without acrylamide, with/without bacteria) was performed in six replicates for each timepoint.

#### 2.6.3. The Evaluation of Acrylamide Degradation by *Lactobacillus acidophilus* LA-5 in Real Solution

In the last part of our experiment, we checked whether the ability to degrade acrylamide was maintained under the actual conditions of the bacteria’s presence, i.e., in milk. Therefore, an experiment was conducted, similarly to the one in Section 2.6.1, except that milk (UHT, 2% of fat content, purchased in a local store, the initial pH was 6.5) was used instead of the MRD medium. The experiment was performed in six replicates.

### 2.7. The Determination of Acrylamide Concentration Changes in Samples

#### 2.7.1. The Preparation of Internal Standard (ITS) Solution

An total of 23.4 mg of the internal standard, methacrylamide (MAA), was weighed, transferred to a 100 mL volumetric flask, and made up with ultrapure water (Water System, Ultrapure, Millipore^®^, Simplicity with UV; water resistivity 18.2 MΩ·cm at 25 °C). A total of 0.25 mL of the internal standard solution was added for each 10 mL sample or standard, which corresponded to a concentration of 10 mg MAA/L. An ITS solution was freshly prepared before each experiment.

#### 2.7.2. The Determination of Acrylamide Concentration by Gas Chromatography (GC)

A 0.2 g measurement of activated charcoal was added to a 15 mL test tube and mixed with 0.2 g of silica gel, followed by the addition of 10 mL of the sample and 0.25 mL of ITS. The samples were then thoroughly mixed and, after 10 min, centrifuged (3000× *g* for 3 min). The supernatant was decanted, and 2 mL of ethyl acetate was added to a pellet. After thorough mixing, 0.8 g of anhydrous sodium sulphate was added, and the sample was centrifuged again using the same parameters. The solvent layer was gently transferred to a 2 mL vial and analysed by gas chromatography (GC-MS Hewlett Packard 5890 Series II with FID detector, range 10,000, rate 10 Hz, autosampler HT 3000A). The chromatographic column used in the study was a Rxi-624 Sil MS column of 30 m length, ID 0.53, film thickness 3.0 µm (Restek Pure Chromatography). The volume of the injected sample was 1 µL. The carrier gas (He) flow was 3 mL/min. The temperature program was as follows: 110 °C for 15 min; 10 °C/min up to 160 °C; 50 °C/min up to 200 °C; 200 °C for 5 min. The temperature of the dispenser and detector was 250 °C. The retention time for acrylamide was 11.1 min, and for MAA the time was 13.0 min. LOD and LOQ for the determination of acrylamide in water solution were estimated via the blank approach at 0.3 mg/L and 1.0 mg/L, respectively.

The results were compiled using the software Clarity 7.X. Final acrylamide concentration in samples was calculated on the basis of calibration curves (made separately for each kind of medium) using an internal standard to correct for the loss of analyte during sample preparation and analysis.

### 2.8. The Impact of Acrylamide on Microbial Metabolism

The aim of this part of the study was to evaluate how acrylamide influenced the metabolism of *Lactobacillus acidophilus* LA-5. For this purpose, the level of lactose, and glucose (the product of lactose hydrolysis), as well as the main metabolites of lactic acid fermentation pathways (lactic acid, acetic acid), were determined using high-performance liquid chromatography (HPLC).

In brief, 1 mL of acrylamide “Stock” solution was added (final concentrations of 7.5, 15, 30, or 100 µg/mL) to tubes containing 8mL of milk (the same as in Section 2.6.2) and inoculated with 1 mL of bacteria suspension (containing 10^7^ cfu/mL). At the beginning of the experiment (time 0 h), and after 24 h and 48 h of incubation at 37 °C, the contents of the samples were centrifuged at 3000× *g* for 3 min (conditions as described above) and the supernatants were collected for further analysis. The experiment was performed in 6 replicates.

#### 2.8.1. Sample Preparation for HPLC Analyses

The analysed sample (0.5 mL) was put into a volumetric flask with a capacity of 25 mL, and 10 mL of a mixture of 95% EtOH and water (1:1) was added and mixed. Then, 0.5 mL of Carrez’s Reagents I and II were added to the flask and mixed. Next, 5 mL of acetonitrile was added, and the total volume was made up to 25 mL with a mixture of 95% EtOH and water (1:1). After 1 h, the solution was filtered and injected on the HPLC column.

#### 2.8.2. The Determination of Sugars Content by HPLC

The content of lactose and glucose was determined by the HPLC method (as described by Tarko et al. 2021 [15]), using the Shimadzu (Kyoto, Japan) NEXERA XR apparatus with an RF-20A refractometric detector. Quantitative determinations were made with the use of standard curves prepared for appropriate sugar standards: glucose and lactose.

#### 2.8.3. The Determination of Lactic and Acetic Acids Content by HPLC

The content of lactic and acetic acids was determined by the HPLC method using the chromatograph mentioned above. Separation was conducted with a Rezex 300 × 7 mm Phenomenex column, thermostated at 40 °C. The mobile phase was an aqueous solution of sulphuric acid (0.005 mol/L) and the elution was made using the isocratic program (0.3 mL/min, 35 min). The quantitative assessment of acids was made with the use of standard curves prepared for appropriate standards: acetic acid and lactic acid.

### 2.9. Statistical Analysis

All experiments described in the current paper were carried out in six replicates, and the results are shown as arithmetical mean ± standard deviation (SD). The Kolmogorov–Smirnov test was used to test the normality of distribution, while Bartlett’s test to verify the hypothesis of the homogeneity of variances. Depending on their results, either the ANOVA with Tukey–Kramer post hoc test or the Kruskal–Wallis (non-parametric) test with Dunn’s test were performed (In Stat, version 3.01, Graph Pad Software, La Jolla, CA, USA).

A *p*-value < 0.05 was considered statistically significant.

## 3. Results and Discussion

### 3.1. Acrylamide Degradation in Carbon- and Nitrogen-Limited Conditions

Table 1 shows the changes in acrylamide concentration caused by the probiotic strain *Lactobacillus acidophilus* LA-5 (the difference between samples without bacteria and with bacteria) for variants in initial acrylamide dose and incubation time. Statistically significant differences were observed at acrylamide concentrations of 15 and 100 µg/mL, and were time-dependent. In the study, it was confirmed that the tested strain was able to decrease the acrylamide concentration in the environment when carbon- and nitrogen-sources are lacking.

In our previous paper [13], it was reported that ammonia only was released in samples with acrylamide in the presence of *L. acidophilus* LA-5. Based on the previous and present results, it was confirmed that *L. acidophilus* LA-5 can degrade the acrylamide and is most probably conducted by the synthesis of amidase, which releases an ammonia from the acrylamide molecule. The enhanced bacterial growth in such samples proved that bacteria could use the released ammonia as a nitrogen source, while the rest of the molecule could be used as a carbon source, in conditions where other available sources are limited. However, it is possible that, in samples with a low acrylamide level (e.g., 7.5 µg/mL), the chances for amidase to meet the acrylamide molecule (and finally carry out the enzymatic reaction of its degradation) was too low, so the majority of AA molecules were not decomposed. The first statistically significant differences were reported for 15 µg AA/mL (in Table 1).

According to Wampler and Ensign [16], acrylamide can be decomposed to ammonia (a nitrogen source) and acrylic acid (a carbon source), which might be then transformed under various, aerobic or anaerobic, conditions to β-hydroxypropionate, which is further oxidised to CO_2_, or to acetate, propionate, lactate, or acrylyl-CoA. However, all previous studies proving acrylamide decomposition by microbial amidases and the release of ammonia and acrylic acid, had not tested LAB species [4,17].

We believe that acrylamide was decomposed and utilized in other samples (30 and 100 µg/mL) as well, but that other mechanisms were likely to overlap with the results. In all samples tested in the study, peptone components were first utilized by bacteria. Only when they had finished could the bacteria utilize another (additional) source of carbon and nitrogen, in this case an acrylamide (after switching their metabolism) and still proliferate (which we proved in our previous study [15]). The higher the AA concentration, the higher the proliferation of *L. acidophilus* LA-5, but acrylamide concentrations of 15 or 30 µg AA/mL were probably still too low to maintain a living and metabolically active culture for a long time. It has been postulated that, in such cultures, the growth-limiting factor seems to be the lack of an available carbon source [15]. It is likely that the higher the number of bacterial cells in a tube, the higher the number that die and undergo cell lysis. This means that a higher number of bacterial cells (in medium with a higher AA) will result in an increase in the absolute number of dead cells (even if the percentage of dying cells remains unchanged) and, as a consequence, more components from lysed cells would be released into the medium. Ammonia (a nitrogen source) can also be produced by bacteria from cell compounds released to the medium from bacteria after cell lysis (e.g., from amino acids) [18], but the growth-limiting factor will still be a carbon source. Therefore, considering the result, where, in samples with 100 µg AA/mL, the changes start to be significant, it can be supposed that additional mechanisms were involved in the enhanced degradation of acrylamide; for example, the induction of bacterial response and adaptation to environmental stress. It is likely that only the AA concentration of 100 µg/mL (but not the lower ones) was able to induce such a response. This process can explain why some bacteria still survived and could actively metabolize the acrylamide.

The acrylamide concentration range was chosen based on the acrylamide levels in food reported in the literature [6,19]. Although the concentration of 100 µg/mL is higher than the possible level reached in food products or in the human gastrointestinal tract, we chose to observe if such a high amount of acrylamide matters or has a toxic impact on bacteria. No harmful influence of high acrylamide doses was observed.

### 3.2. The Impact of Milk Components on Acrylamide Degradation

When one of the basic milk components (casein, lactose, or milk fat) was added to the low-carbon and low-nitrogen medium (MRD), the ability of tested bacteria to acrylamide degradation disappeared, as the changes in acrylamide concentration did not statistically differ in comparison to the initial level. The same situation was observed when bacteria were directly inoculated to cow’s milk, independently from the acrylamide concentration or incubation time. This means that when casein, lactose or milk fat are present in the medium at concentrations usually found in cow’s milk, or bacteria were inoculated to the whole cow’s milk, an acrylamide was no longer a competitive source of nitrogen or carbon, and bacteria were preferentially used compounds for the metabolism to which they were evolutionarily adapted. Only in a poor environment, lacking easily available nutrients, was the probiotic strain of lactic acid bacteria able to degrade acrylamide, as confirmed by a statistically significant decrease in its concentration.

To date, proven and described methods of acrylamide mitigation consisted either in the prevention of its synthesis in food or in the removal of the already formed acrylamide [3]. Various bacteria or their enzymes (L-asparaginase) were used to catalyse the conversion of L-asparagine to L-aspartic acid, therefore removing the substrate of acrylamide synthesis from the food and preventing acrylamide formation [20,21,22]. Some studies have also revealed that LAB can reduce the acrylamide concentration, due to its binding to the bacterial cell wall [6,23,24]. Serrano-Niño et al. [24] demonstrated that tested *Lactobacillus* strains were able to remove acrylamide in phosphate-buffered saline by irreversible binding, and this was a strain- and concentration-dependent process. However, the acrylamide binding was inhibited when its concentration increased to 10 µg/mL. In our study, acrylamide degradation was observed even at a 10-fold higher concentration. The AA degradation ability of amidases has already been confirmed in some soil or water bacteria, as well as in some species that are naturally present in the human gastrointestinal tract or that are delivered with food [5]. This study proved that *L. acidophilus* LA-5, a probiotic strain of LAB, can also degrade acrylamide and decrease its concentration. To exclude the possibility that the decrease in acrylamide concentration, as reported in the study, was caused by its binding to the cell wall of bacteria, an additional experiment was performed. This was similar to that which was previously described, but samples were inoculated with a 10-fold higher number of bacteria cells. However, a decrease in AA concentration could only be observed in a poor environment (MRD medium). In environments rich in easily digested and assimilated nutrients (MRD supplemented with milk components, the whole cow’s milk), no decrease in AA concentration was observed, even with more bacteria. Therefore, one can conclude that the changes in AA concentration could not be caused by its adsorption. Moreover, as was reported before [13], ammonia was only released in the simultaneous presence of bacteria and AA, which suggests that AA degradation was dependent on bacteria activity and amidase synthesis. Differences in pH between samples without acrylamide or with various AA concentrations were not noted or they were not statistically significant when measured at a particular timepoint (data not present). Therefore, the impact of pH changes on AA concentration in samples could be excluded.

### 3.3. The Impact of Acrylamide on Microbial Metabolism–Analysis of Produced Metabolites

As mentioned above, in an environment rich in nutrients, *L. acidophilus* LA-5 did not degrade the AA molecule. Therefore, we checked whether the metabolism of a tested probiotic strain was influenced by the presence of AA in milk.

It is obvious that LAB is responsible for the sensory quality of milk-fermented products, and three main pathways are involved: glycolysis (fermentation of sugars), proteolysis (degradation of proteins) and lipolysis (degradation of fat). Lactic acid is the main product generated during the lactose metabolism. However, some part of a pyruvate, which is the intermediate in this reaction, can alternatively be converted to acetic acid, acetaldehyde, diacetyl, or acetoin, depending on the bacteria species [25]. The lactose metabolism depends on the LAB species, pH, and medium composition; the lactose can be metabolised by the homo- or heterofermentative pathway. For example, *Lactobacillus bulgaricus* and *L. acidophilus* are homo-fermentative LAB, with the glycolytic pathway as a major metabolic process, and lactic acid is the main product of sugar metabolism. For every mole of monosaccharides entering the cells through phosphorylation and enzyme reaction, two moles of lactic acid are generated [26].

In experiments, it was confirmed that glucose and lactose were used for the growth and metabolism of *L. acidophilus* LA-5. Glucose was not detected in any sample of tested milk after fermentation (data are not shown), indicating that this is the best carbon source for the examined bacteria. The lactose concentration decreased within the incubation time (Figure 1), and the dose of acrylamide had a slight impact on the ratio of lactose metabolism. The only statistically significant differences were noted at the first 24 h in samples where AA was present at a concentration of 7.5 µg/mL. In this case, the amount of degraded lactose was slightly lower than that at higher AA concentrations.

*L. acidophilus* LA-5 is a homofermentative bacteria, so only lactic acids should be produced during milk fermentation [25]. However, it was proved that, under stress conditions (for example, when the carbon source is limited or when carbon sources other than glucose are present), some homofermentative microorganisms can produce other organic acids due to their changed metabolic pathways [25]. Kim et al. [27] proved that *L. acidophilus* could have an adaptive response to stress and that this adaptive response to one kind of stress could provide a cross-protection against other stress tests. Lactic acid bacteria, especially probiotics strains, are of high commercial importance. Their stress physiology and response to the stress factors during various steps of production, such as the storage of fermented foods as well as during the passage through the gastrointestinal tract after consumption, were studied in detail [28,29].

It has been reported that the highest used AA concentration (100 μg/mL) had a statistically significant impact on the production of lactic acid by *L. acidophilus* LA-5, and its concentration was 2–3 times higher than that in other samples (Figure 2). It is interesting that AA, independently of its concentration, changed the metabolism pathways of LA-5 and acetic acid production is reduced (Figure 3) when compared with samples without AA, although the differences were not statistically significant.

The presence of acetic acid in the media after fermentation is highly differentiated among *Lactobacillus* species, or even strains, as it can result from various biochemical pathways. For example, it can be the product of lactic acid degradation, or the result of the citrate metabolism, or it can originate from the heterofermentative pathway [26,30]. The results of Hickey et al. [30] indicated that *L. acidophilus* oxidised pyruvate via the pyruvate oxidase system, and acetyl phosphate was an intermediate in the reaction of pyruvate conversion to acetate.

However, unlike other *Lactobacillus* species, *L. acidophilus* did not utilise citrate under the conditions used. Meremäe et al. [31] proved that *L. acidophilus* ATCC 4356 was the only probiotic strain that was able to totally inhibit the growth and survival of *Campylobacter jejuni* when was used in the combination of 1% inulin or 1% oligofructose. Authors explained this by the fact that a tested strain of *L. acidophilus*, in the presence of prebiotics, produced significantly more lactic and acetic acids than any other probiotic bacteria in their study.

Bacteria have various sensors that allow them to monitor the environment. This means that bacteria can regulate their physiology in a way that is necessary to survive the changes. The changes in the carbohydrate metabolism, as well as in glycolysis and fate of pyruvate were observed in respiratory and environmentally stressed lactic acid bacteria [28]. Stress factors can include low-pH conditions, carbohydrate starvation, oxidative stress, osmotic stress, too-high or too-low temperature, and many more [29]. Modifying the enzyme synthesis and changing the obtained metabolites enables lactic acid bacteria to adapt their metabolism to the new environment and to new carbon sources. For example, under low-pH conditions, some lactobacilli can increase the activity of pyruvate kinase, which may accelerate the depletion of fructose-1,6-bisphosphate. This relieves CcpA repression and allows for the use of alternative carbon sources [32]. It was proven that nitrogen metabolism is regulated in LAB, mainly via GlnR and CodY [27]. Interestingly, the GlnR gene was detected in each of the sequenced genomes of LAB, while gene for CodY was only detected in strains belonging to the *Lactococcus*, *Streptococcus*, and *Enterococcus* genera. It was shown that CodY activates the expression of a germination-specific amidase in *C. difficile* [33]. The fast induction of various other genes responsible for LAB survival under stress conditions can also be regulated by an alternative sigma factor [29].

## 4. Conclusions

Our results proved that probiotic strain *L. acidophilus* LA-5 can degrade acrylamide and decrease its concentration in the environment. However, this ability was only reported for the poor nutrient medium, where other, easily digestible nutrients are lacking. In an environment that was rich in nutrients, LA5 did not use its ability to degrade acrylamide. This means that the reduction in AA concentration will not occur in yoghurts or other fermented milk beverages with additives that contain AA (muesli, roasted nuts, crackers, roasted coffee, etc.). In this kind of food, other easily available nutrients are present, and bacteria would choose them before AA.

Our results have also revealed that the presence of acrylamide was a stress factor, which modulated the metabolism of *L. acidophilus* LA-5. This means that the acrylamide that was present or unintentionally added to food (e.g., with additives) not only exerts a negative impact on consumer health, but can also change the sensory parameters of fermented food, causing financial losses.

## Figures and Tables

**Figure 1 foods-11-00365-f001:**
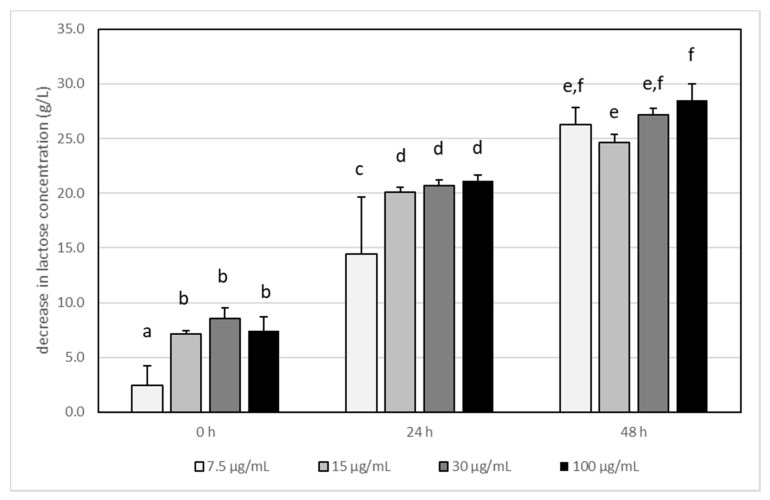
The influence of acrylamide (AA) concentration on lactose degradation by *Lactobacillus acidophilus* LA-5 in milk during incubation at 37 °C. Bars represent the differences (arithmetic mean ± SD, *n* = 6) in lactose concentration between control (i.e., milk without AA) and test sample (milk with adequate AA concentration) The means with different letters (a–f) differ from each other at the level of *p* < 0.05 (ANOVA with Tukey–Kramer post hoc test).

**Figure 2 foods-11-00365-f002:**
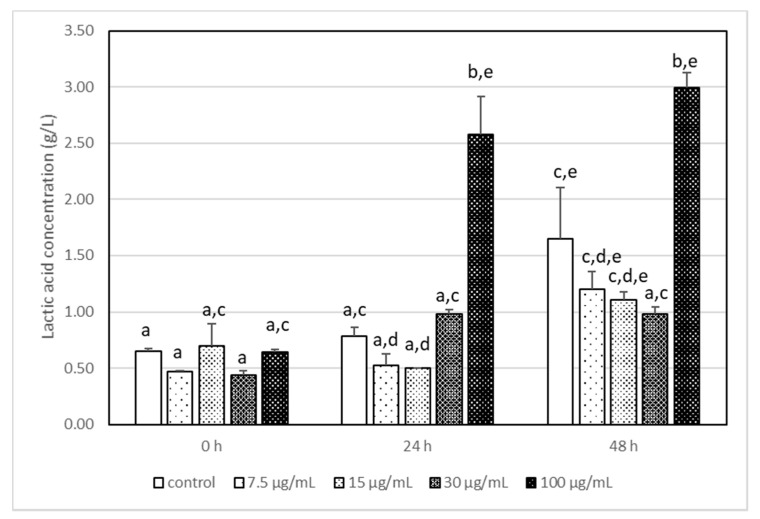
The changes in the lactic acid concentration in milk after incubation at 37 °C with LA-5 in the presence of various acrylamide concentrations (7.5–100 µg/mL). The means with different letters (a–e) differ from each other at the level of *p* < 0.05. Kruskal–Wallis test with Dunn’s post hoc test were performed.

**Figure 3 foods-11-00365-f003:**
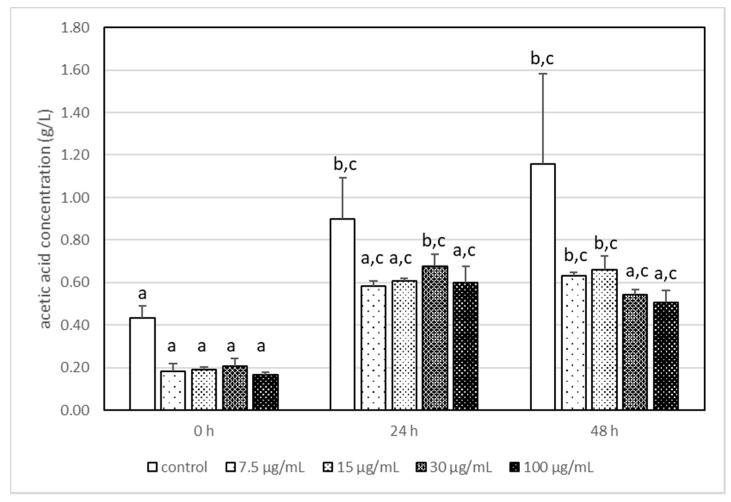
The changes in the acetic acid concentration in milk after incubation at 37 °C with LA-5 in the presence of various acrylamide concentrations (7.5–100 µg/mL). The means with different letters (a–c) differ from each other at the level of *p* < 0.05. Kruskal–Wallis test with Dunn’s post hoc test were performed.

**Table 1 foods-11-00365-t001:** The acrylamide degradation caused by *Lactobacillus acidophilus* LA-5 added to the MRD medium containing various concentrations of acrylamide during incubation (*n* = 6).

Time (h)	Initial Acrylamide Concentration (µg/mL)	Changes in Acrylamide Concentration (Positive Control vs. Test Sample)
0	7.5	0.1 ± 0.28 ^a^
24		0.2 ± 0.32 ^a^
48		0.4 ± 0.16 ^a^
0	15.0	4.4 ± 0.40 ^a^
24		5.1 ± 0.22 ^b^
48		5.7 ± 0.22 ^c^
0	30.0	0.2 ± 0,59 ^a^
24		0.3 ± 0,18 ^a^
48		0.7 ± 0.92 ^a^
0	100.0	0.6 ± 0.39 ^a^
24		4.6 ± 0.89 ^b^
48		7.0 ± 2.25 ^c^

Values (arithmetic mean ± SD) represent the difference in AA concentration measured after incubation time in positive control (medium with AA without bacteria) and the AA concentration in the test sample (medium with the same dose of AA and with bacteria). a, b, c —the same letters next to the means indicate the lack of statistical differences (at *p* < 0.05).

## Data Availability

Not applicable.

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
