# Peer review of "The Acrylamide Degradation by Probiotic Strain Lactobacillus acidophilus LA-5"

_foods, 2022, doi:10.3390/foods11030365_

Round 1
Reviewer 1 Report
In the manuscript aa probiotic strais of Lactabacillus is firstly used for degradation of acrylamide level of model food system. The results of study are very impressive since the reducing acrylamide level is paid attention in food science. The organisation of study and given of data are effective. Therefore, the listed minor revisions should be corrected.
-In the introduction part, the emphasise on gluten should be removed. Because Maillard reactions occur in every food matrix even there is no gluten present.
-In section 2.5. What was the rehydration procedure
-In section 2.9.2, The reference of acrylamide determination method should be given. Because most of the methods are used bromine derivatisation in GC
Author Response
Our answer to the reviewer comments are in attached file.

Reviewer 2 Report
The objective of this study was to evaluate the ability of probiotic bacteria strain, namely, Lactobacillus acidophilus LA-5 (LA5), to degrade acrylamide and reduce its concentration in foodstuff. The manuscript is well written and organized. However, some major concerns should be clarified.
The manuscript needs to be checked by English native for some weak or misspelling sentences.
L269: clarify the types of organic acids?
L288: Please clarify why the significant differences were observed at acrylamide concentration 15 and 100 µg/mL only? Why not 7.5 or 30 µg/mL?
The main findings of this study was in the presence of casein, lactose, milk fat or in whole cow’s milk this ability disappeared. This means that Lactobacillus acidophilus LA-5 will not reduce the acrylamide levels in food products particularly fermented products which are usually rich in simple carbohydrates? So the question raised what are the benefits of addition this strain in case of acrylamide?
Write an application statement for these findings?
Table 1: the values are not clear? Are they the reduction in acrylamide concentration?
The populations (numbers) of Lactobacillus acidophilus LA-5 in MRD and milk could be added. Further, have you checked out the presence of natural microflora in UHT milk used?
Figure 1: why there were significant differences in the reduction levels of lactose at zero (0) h between the different concentrations of AA? This should be equal value or it seems that presence of higher concentration of AA block detection of lactose by the adapted method?
Figure 1-3: show the incubation temperature in the title of figure?
Figure 1-3: be consistent in x-axis, it is better to AA concentration in figure 2-3.
Figure 1: positive inoculated control should be added (medium inoculated with bacteria but in the absence of AA)
Have the investigators measured the pH of medium before and after incubation or fermentation?
Minor corrections:
L17: add "by" before " significantly"
L24-25: "probably cancerogenic" compound
L27: delete "the so called"
L49: Italicize: in vitro, ex vivo, in vivo
L71-75: have these studies used food as a medium for probiotics or just used culture media? Please clarify this to show the difference in the current study?
L95: add city before Poland "by Chempur (Poland)"
L196: Italicize: Lactobacillus acidophilus
L442: the poor nutrient medium
Author Response

(The authors gave the same response as above.)

Round 2
Reviewer 2 Report
Th authors revised the manuscript according to the suggestions and the manuscript is improved and can be accepted in the current form.
Author Response
Thank you very much :-)